# Computational Complexity of Detecting Proximity to Losslessly Compressible Neural Network Parameters

## Abstract

To better understand complexity in neural networks, we theoretically investigate the idealised phenomenon of lossless network compressibility, whereby an identical function can be implemented with a smaller network. We give an efficient formal algorithm for optimal lossless compression in the setting of single-hidden-layer hyperbolic tangent networks. To measure lossless compressibility, we define the *rank* of a parameter as the minimum number of hidden units required to implement the same function. Losslessly compressible parameters are atypical, but their existence has implications for nearby parameters. We define the *proximate rank* of a parameter as the rank of the most compressible parameter within a small $L^\infty$ neighbourhood. Unfortunately, detecting nearby losslessly compressible parameters is not so easy: we show that bounding the proximate rank is an $\mathcal{NP}$-complete problem, using a reduction from Boolean satisfiability via a novel abstract clustering problem involving covering points with small squares. These results underscore the computational complexity of measuring neural network complexity, laying a foundation for future theoretical and empirical work in this direction.

## 1 Introduction

Learned neural networks often generalise well, depite the excessive expressive capacity of their architectures (Zhang et al., 2017, 2021). Moreover, learned neural networks are often *approximately compressible*, in that smaller networks can be found implementing similar functions (via, e.g., model distillation, Buciluǎ et al., 2006; Hinton et al., 2014; see, e.g., Sanh et al., 2019 for a large-scale example). In other words, learned neural networks are often simpler than they might seem. Advancing our understanding of neural network complexity is key to understanding deep learning.

We propose studying the idealised phenomenon of *lossless compressibility*, whereby an *identical* function can be implemented with a smaller network.[1] Classical functional equivalence results imply that, in many architectures, almost all parameters are incompressible in this lossless, unit-based sense (e.g., Sussmann, 1992; Chen et al., 1993; Fefferman, 1994; Phuong and Lampert, 2020). However, these results specifically exclude measure zero sets of parameters with more complex functional equivalence classes (Anonymous, 2023), some of which are losslessly compressible.

We argue that, despite their atypicality, losslessly compressible parameters may be highly relevant to deep learning. The learning process exerts a non-random selection pressure on parameters, and losslessly compressible parameters are appealing solutions due to parsimony. Moreover, losslessly compressible parameters are a source of information singularities (cf. Fukumizu, 1996), highly relevant to statistical theories of deep learning (Watanabe, 2009; Wei et al., 2022).

---

[1]We measure the size of a neural network for compression purposes by the number of units. Other conventions are possible, such as counting the number of weights, or the description length of specific weights.

Submitted to 37th Conference on Neural Information Processing Systems (NeurIPS 2023). Do not distribute.

Even if losslessly compressible parameters themselves are rare, their aggregate parametric neighbourhoods have nonzero measure. These neighbourhoods have a rich structure that reaches throughout the parameter space (Anonymous, 2023). The parameters in these neighbourhoods implement similar functions to their losslessly compressible neighbours, so they are necessarily approximately compressible. Their proximity to information singularities also has implications for local learning dynamics (Amari et al., 2006; Wei et al., 2008; Cousseau et al., 2008; Amari et al., 2018).

In this paper, we study losslessly compressible parameters and their neighbours in the setting of single-hidden-layer hyperbolic tangent networks. While this architecture is not immediately relevant to modern deep learning, parts of the theory are generic to feed-forward architecture components. A comprehensive investigation of this simple and concrete case is a first step towards studying more modern architectures. To this end, we offer the following theoretical contributions.

1. In Section 4, we give efficient formal algorithms for optimal lossless compression of single-hidden-layer hyperbolic tangent networks, and for computing the *rank* of a parameters—the minimum number of hidden units required to implement the same function.

2. In Section 5, we define the *proximate rank*—the rank of the most compressible parameter within a small $L^\infty$ neighbourhood. We give a greedy algorithm for bounding this value.

3. In Section 6, we show that bounding the proximate rank below a given value (that is, detecting proximity to parameters with a given maximum rank), is an $\mathcal{NP}$-complete decision problem. The proof involves a reduction from Boolean satisfiability via a novel abstract decision problem involving clustering points in the plane into small squares.

These results underscore the computational complexity of measuring neural network complexity: we show that while lossless network compression is easy, detecting highly-compressible networks near a given parameter can be very hard indeed (embedding any computational problem in $\mathcal{NP}$). Our contributions lay a foundation for future theoretical and empirical work detecting proximity to losslessly compressible parameters in learned networks using modern architectures. In Section 7, we discuss these research directions, and limitations of the lossless compressibility framework.

## 2   Related work[2]

Two neural network parameters are *functionally equivalent* if they implement the same function. In single-hidden-layer hyperbolic tangent networks, Sussmann (1992) showed that, for almost all parameters, two parameters are functionally equivalent if and only if they are related by simple operations of exchanging and negating the weights of hidden units. Similar operations have been found for various architectures, including different nonlinearities (e.g., Albertini et al., 1993; Kůrková and Kainen, 1994), multiple hidden layers (e.g., Fefferman and Markel, 1993; Fefferman, 1994; Phuong and Lampert, 2020), and more complex connection graphs (Vlačić and Bölcskei, 2021, 2022).

Lossless compressibility requires functionally equivalent parameters in smaller architectures. In all architectures where functional equivalence has been studied (cf. above), the simple operations identified do not change the number of units. However, all of these studies explicitly exclude from consideration certain measure zero subsets of parameters with richer functional equivalence classes. The clearest example of this crucial assumption comes from Sussmann (1992), whose result holds exactly for "minimal networks" (in our parlance, losslessly incompressible networks).

Anonymous (2023) relaxes this assumption, studying functional equivalence for non-minimal single-hidden-layer hyperbolic tangent networks. Anonymous (2023) gives an algorithm for finding canonical equivalent parameters using various opportunities for eliminating or merging redundant units.[3] This algorithm implements optimal lossless compression as a side-effect. We give a more direct and efficient lossless compression algorithm using similar techniques.

Beyond *lossless* compression, there is a significant empirical literature on approximate compressibility and compression techniques in neural networks, including via network pruning, weight quantisation, and student–teacher learning (or model distillation). Approximate compressibility has also

---

[2]We discuss related work in computational complexity throughout the paper (Section 6 and Appendix B).

[3]Patterns of unit redundancies have also been studied by Fukumizu and Amari (2000), Fukumizu et al. (2019), and Şimşek et al. (2021), though from a dual perspective of cataloguing various ways of *adding* hidden units to a neural network while preserving the implemented function (lossless *expansion*, so to speak).

82 been proposed as a learning objective (see, e.g., Hinton and van Camp, 1993; Aytekin et al., 2019)
83 and used as a basis for generalisation bounds (Suzuki et al., 2020a,b). For an overview, see Cheng
84 et al. (2018, 2020) or Choudhary et al. (2020). Of particular interest is a recent empirical study of
85 network pruning from Casper et al. (2021), who, while investigating the structure of learned neu-
86 ral networks, found many instances of units with weak or correlated outputs. Casper et al. (2021)
87 found that these units could be removed without a large effect on performance, using elimination
88 and merging operations bearing a striking resemblance to those discussed by Anonymous (2023).

# 3 Preliminaries

We consider a family of fully-connected, feed-forward neural network architectures with one input unit, one biased output unit, and one hidden layer of $h \in \mathbb{N}$ biased hidden units with the hyperbolic tangent nonlinearity $\tanh(z) = (e^z - e^{-z})/(e^z + e^{-z})$. The weights and biases of the network are encoded in a parameter vector in the format $w = (a_1, b_1, c_1, \ldots, a_h, b_h, c_h, d) \in \mathcal{W}_h = \mathbb{R}^{3h+1}$, where for each hidden unit $i = 1, \ldots, h$ there is an *outgoing weight* $a_i \in \mathbb{R}$, an *incoming weight* $b_i \in \mathbb{R}$, and a *bias* $c_i \in \mathbb{R}$; and $d \in \mathbb{R}$ is the *output unit bias*. Thus each parameter $w \in \mathcal{W}_h$ indexes a mathematical function $f_w : \mathbb{R} \to \mathbb{R}$ such that $f_w(x) = d + \sum_{i=1}^{h} a_i \tanh(b_i x + c_i)$. All of our results generalise to networks with multi-dimensional inputs and outputs.

Two parameters $w \in \mathcal{W}_h, w' \in \mathcal{W}_{h'}$ are *functionally equivalent* if $f_w = f_{w'}$ as functions on $\mathbb{R}$ ($\forall x \in \mathbb{R}, f_w(x) = f_{w'}(x)$). A parameter $w \in \mathcal{W}_h$ is *(losslessly) compressible* (or *non-minimal*) if and only if $w$ is functionally equivalent to some $w' \in \mathcal{W}_{h'}$ with fewer hidden units $h' < h$ (otherwise, $w$ is *incompressible* or *minimal*). Sussmann (1992) showed that a simple condition, *reducibility,* is necessary and sufficient for lossless compressibility. A parameter $(a_1, b_1, c_1, \ldots, a_h, b_h, c_h, d) \in \mathcal{W}_h$ is *reducible* if and only if it satisfies any of the following *reducibility conditions*:

(i) $a_i = 0$ for some $i$, or

(ii) $b_i = 0$ for some $i$, or

(iii) $(b_i, c_i) = (b_j, c_j)$ for some $i \neq j$, or

(iv) $(b_i, c_i) = (-b_j, -c_j)$ for some $i \neq j$.

Each reducibility condition suggests a simple operation to remove a hidden unit while preserving the function (Sussmann, 1992; Anonymous, 2023): (i) units with zero outgoing weight do not contribute to the function; (ii) units with zero incoming weight contribute a constant that can be incorporated into the output bias; and (iii), (iv) unit pairs with identical (negative) incoming weight and bias contribute in proportion (since the hyperbolic tangent is odd), and can be merged into a single unit with the sum (difference) of their outgoing weights.

Define the *uniform norm* (or $L^\infty$ *norm*) of a vector $v \in \mathbb{R}^p$ as $\|v\|_\infty = \max_{i=1}^p \mathrm{abs}(v_i)$, the largest absolute component of $v$. Define the *uniform distance* between $v$ and $u \in \mathbb{R}^p$ as $\|u - v\|_\infty$. Given a positive scalar $\varepsilon \in \mathbb{R}^+$, define the *closed uniform neighbourhood of $v$ with radius $\varepsilon$, $\bar{B}_\infty(v; \varepsilon)$,* as the set of vectors of distance at most $\varepsilon$ from $v$: $\bar{B}_\infty(v; \varepsilon) = \{ u \in \mathbb{R}^p : \|u - v\|_\infty \leq \varepsilon \}$.

A *decision problem*[4] is a tuple $(I, J)$ where $I$ is a set of *instances* and $J \subseteq I$ is a subset of *affirmative instances*. A *solution* is a deterministic algorithm that determines if any given instance $i \in I$ is affirmative ($i \in J$). A *reduction* from one decision problem $X = (I, J)$ to another $Y = (I', J')$ is a deterministic polytime algorithm implementing a mapping $\varphi : I \to I'$ such that $\varphi(i) \in J' \Leftrightarrow i \in J$. If such a reduction exists, say $X$ *is reducible*[5] *to* $Y$ and write $X \to Y$. Reducibility is transitive.

$\mathcal{P}$ is the class of decision problems with polytime solutions (polynomial in the instance size). $\mathcal{NP}$ is the class of decision problems for which a deterministic polytime algorithm can verify affirmative instances given a certificate. A decision problem $Y$ is $\mathcal{NP}$-*hard* if all problems in $\mathcal{NP}$ are reducible to $Y$ ($\forall X \in \mathcal{NP}, X \to Y$). $Y$ is $\mathcal{NP}$-*complete* if $Y \in \mathcal{NP}$ and $Y$ is $\mathcal{NP}$-hard. Boolean satisfiability is a well-known $\mathcal{NP}$-complete decision problem (Cook, 1971; Levin, 1973; see also Garey and Johnson, 1979). $\mathcal{NP}$-complete decision problems have no known polytime exact solutions.

---

[4]We informally review several basic notions from computational complexity theory. Consult Garey and Johnson (1979) for a rigorous introduction (in terms of formal languages, encodings, and Turing machines).

[5]Context should suffice to distinguish reducibility *between decision problems* and *of network parameters*.

## 4 Lossless compression and rank

We consider the problem of lossless neural network compression: finding, given a compressible parameter, a functionally equivalent but incompressible parameter. The following algorithm solves this problem by eliminating units meeting reducibility conditions (i) and (ii), and merging unit pairs meeting reducibility conditions (iii) and (iv) in ways preserving functional equivalence.

**Algorithm 4.1** (Lossless neural network compression). Given $h \in \mathbb{N}$, proceed:

1: **procedure** COMPRESS($w = (a_1, b_1, c_1, \ldots, a_h, b_h, c_h, d) \in \mathcal{W}_h$)
2:     ▷ *Stage 1: Eliminate units with incoming weight zero (incorporate into new output bias $\delta$)* ◁
3:     $I \leftarrow \{ i \in \{1, \ldots, h\} : b_i \neq 0 \}$
4:     $\delta \leftarrow d + \sum_{i \notin I} \tanh(c_i) \cdot a_i$
5:     ▷ *Stage 2: Partition and merge remaining units by incoming weight and bias* ◁
6:     $\Pi_1, \ldots, \Pi_J \leftarrow$ partition $I$ by the value of $\mathrm{sign}(b_i) \cdot (b_i, c_i)$
7:     **for** $j \leftarrow 1, \ldots, J$ **do**
8:        $\alpha_j \leftarrow \sum_{i \in \Pi_j} \mathrm{sign}(b_i) \cdot a_i$
9:        $\beta_j, \gamma_j \leftarrow \mathrm{sign}(b_{\min \Pi_j}) \cdot (b_{\min \Pi_j}, c_{\min \Pi_j})$
10:    **end for**
11:    ▷ *Stage 3: Eliminate merged units with outgoing weight zero* ◁
12:    $k_1, \ldots, k_r \leftarrow \{ j \in \{1, \ldots, J\} : \alpha_j \neq 0 \}$
13:    ▷ *Construct a new parameter with the remaining merged units* ◁
14:    **return** $(\alpha_{k_1}, \beta_{k_1}, \gamma_{k_1}, \ldots, \alpha_{k_r}, \beta_{k_r}, \gamma_{k_r}, \delta) \in \mathcal{W}_r$
15: **end procedure**

**Theorem 4.1** (Algorithm 4.1 correctness). *Given $w \in \mathcal{W}_h$, compute $w' = $ COMPRESS$(w) \in \mathcal{W}_r$. (i) $f_{w'} = f_w$, and (ii) $w'$ is incompressible.*

*Proof sketch* (Full proof in Appendix A). For (i), note that units eliminated in Stage 1 contribute a constant $a_i \tanh(c_i)$, units merged in Stage 2 have proportional contributions ($\tanh$ is odd), and merged units eliminated in Stage 3 do not contribute. For (ii), by construction, $w'$ satisfies no reducibility conditions, so $w'$ is not reducible and thus incompressible by Sussmann (1992). ◇

We define the *rank*[6] of a neural network parameter $w \in \mathcal{W}_h$, denoted $\mathrm{rank}(w)$, as the minimum number of hidden units required to implement $f_w$: $\mathrm{rank}(w) = \min \{ h' \in \mathbb{N} : \exists w' \in \mathcal{W}_{h'}; f_w = f_{w'} \}$. The rank is also the number of hidden units in COMPRESS$(w)$, since Algorithm 4.1 produces an incompressible parameter, which is minimal by definition. Computing the rank is therefore a trivial matter of counting the units, after performing lossless compression. The following is a streamlined algorithm, following Algorithm 4.1 but removing steps that don't influence the final count.

**Algorithm 4.2** (Rank of a neural network parameter). Given $h \in \mathbb{N}$, proceed:

1: **procedure** RANK($w = (a_1, b_1, c_1, \ldots, a_h, b_h, c_h, d) \in \mathcal{W}_h$)
2:     ▷ *Stage 1: Identify units with incoming weight nonzero* ◁
3:     $I \leftarrow \{ i \in \{1, \ldots, h\} : b_i \neq 0 \}$
4:     ▷ *Stage 2: Partition and compute outgoing weights for merged units* ◁
5:     $\Pi_1, \ldots, \Pi_J \leftarrow$ partition $I$ by the value of $\mathrm{sign}(b_i) \cdot (b_i, c_i)$
6:     $\alpha_j \leftarrow \sum_{i \in \Pi_j} \mathrm{sign}(b_i) \cdot a_i$ **for** $j \leftarrow 1, \ldots, J$
7:     ▷ *Stage 3: Count merged units with outgoing weight nonzero* ◁
8:     **return** $|\{ j \in \{1, \ldots, J\} : \alpha_j \neq 0 \}|$       ▷ $|S|$ *denotes set cardinality*
9: **end procedure**

**Theorem 4.2** (Algorithm 4.2 correctness). *Given $w \in \mathcal{W}_h$, $\mathrm{rank}(w) = $ RANK$(w)$.*

*Proof.* Let $r$ be the number of hidden units in COMPRESS$(w)$. Then $r = \mathrm{rank}(w)$ by Theorem 4.1. Moreover, comparing Algorithms 4.1 and 4.2, observe RANK$(w) = r$. □

**Remark 4.3.** Both Algorithms 4.1 and 4.2 require $\mathcal{O}(h \log h)$ time if the partitioning step is performed by first sorting the units by lexicographically non-decreasing $\mathrm{sign}(b_i) \cdot (b_i, c_i)$.

---

[6]In the multi-dimensional case, our notion of rank generalises the familiar notion from linear algebra, where the rank of a linear transformation corresponds to the minimum number of hidden units required to implement the transformation with an unbiased linear neural network (cf. Piziak and Odell, 1999). Unlike in the linear case, our non-linear rank is not bound by the input and output dimensionalities.

## 5 Proximity to low-rank parameters

Given a neural network parameter $w \in \mathcal{W}_h$ and a positive radius $\varepsilon \in \mathbb{R}^+$, we define the *proximate rank* of $w$ at radius $\varepsilon$, denoted $\mathrm{prank}_\varepsilon(w)$, as the rank of the lowest-rank parameter within a closed uniform ($L^\infty$) neighbourhood of $w$ with radius $\varepsilon$. That is,

$$\mathrm{prank}_\varepsilon(w) = \min \left\{ \mathrm{rank}(u) \in \mathbb{N} \, : \, u \in \bar{B}_\infty(w; \varepsilon) \right\}.$$

The proximate rank measures the proximity of $w$ to the set of parameters with a given rank bound, that is, sufficiently losslessly compressible parameters.

The following greedy algorithm computes an upper bound on the proximate rank. The algorithm replaces each of the three stages of Algorithm 4.2 with a relaxed version, as follows.

1. Instead of eliminating units with zero incoming weight, eliminate units with *near* zero incoming weight (there is a nearby parameter where these are zero).

2. Instead of partitioning the remaining units by $\mathrm{sign}(b_i) \cdot (b_i, c_i)$, *cluster* them by *nearby* $\mathrm{sign}(b_i) \cdot (b_i, c_i)$ (there is a nearby parameter where they have the same $\mathrm{sign}(b_i) \cdot (b_i, c_i)$).

3. Instead of eliminating merged units with zero outgoing weight, eliminate merged units with *near* zero outgoing weight (there is a nearby parameter where these are zero).

Step (2) is non-trivial, we use a greedy approach, described separately as Algorithm 5.2.

**Algorithm 5.1** (Greedy bound for proximate rank). Given $h \in \mathbb{N}$, proceed:
1: **procedure** BOUND($\varepsilon \in \mathbb{R}^+$, $w = (a_1, b_1, c_1, \ldots, a_h, b_h, c_h, d) \in \mathcal{W}_h$)
2:     ▷ *Stage 1: Identify units with incoming weight not near zero*     ◁
3:     $I \leftarrow \{ i \in \{1, \ldots, h\} \, : \, \mathrm{abs}(b_i) > \varepsilon \}$
4:     ▷ *Stage 2: Compute outgoing weights for nearly-mergeable units*     ◁
5:     $\Pi_1, \ldots, \Pi_J \leftarrow$ APPROXPARTITION($\varepsilon$, $\mathrm{sign}(b_i) \cdot (b_i, c_i)$ **for** $i \in I$)     ▷ *Algorithm 5.2*
6:     $\alpha_j \leftarrow \sum_{i \in \Pi_j} \mathrm{sign}(b_i) \cdot a_i$ **for** $j \leftarrow 1, \ldots, J$
7:     ▷ *Stage 3: Count nearly-mergeable units with outgoing weight not near zero*     ◁
8:     **return** $|\{ j \in \{1, \ldots, J\} \, : \, \mathrm{abs}(\alpha_j) > \varepsilon \cdot |\Pi_j| \}|$     ▷ $|S|$ *denotes set cardinality*
9: **end procedure**

**Algorithm 5.2** (Greedy approximate partition). Given $h \in \mathbb{N}$, proceed:
1: **procedure** APPROXPARTITION($\varepsilon \in \mathbb{R}^+$, $u_1, \ldots, u_h \in \mathbb{R}^2$)
2:     $J \leftarrow 0$
3:     **for** $i \leftarrow 1, \ldots, h$ **do**
4:        **if for some** $j \in \{1, \ldots, J\}$, $\|u_i - v_j\|_\infty \leq \varepsilon$ **then**
5:           $\Pi_j \leftarrow \Pi_j \cup \{i\}$     ▷ *If near a group-starter, join that group.*
6:        **else**
7:           $J, v_{J+1}, \Pi_{J+1} \leftarrow J + 1, u_i, \{i\}$     ▷ *Else, start a new group with this vector.*
8:        **end if**
9:     **end for**
10:    **return** $\Pi_1, \ldots, \Pi_J$
11: **end procedure**

**Theorem 5.1** (Algorithm 5.1 correctness). *For $w \in \mathcal{W}_h$ and $\varepsilon \in \mathbb{R}^+$, $\mathrm{prank}_\varepsilon(w) \leq$ BOUND($\varepsilon, w$).*

*Proof sketch* (Full proof in Appendix A). Trace the algorithm to construct a parameter $u \in \bar{B}_\infty(w; \varepsilon)$ with $\mathrm{rank}(u) =$ BOUND($\varepsilon, w$). During Stage 1, set the nearly-eliminable incoming weights to zero. Use the group-starting vectors $v_1, \ldots, v_J$ from Algorithm 5.2 to construct mergeable incoming weights and biases during Stage 2. During Stage 3, subtract or add a fraction of the merged unit outgoing weight from the outgoing weights of the original units.     ◇

**Remark 5.2.** Both Algorithms 5.1 and 5.2 have worst-case runtime complexity $\mathcal{O}(h^2)$.

**Remark 5.3.** Algorithm 5.1 does *not* compute the proximate rank—merely an upper bound. There may exist a more efficient approximate partition than the one found by Algorithm 5.2. It turns out that this suboptimality is fundamental—computing a smallest approximate partition is $\mathcal{NP}$-hard, and can be reduced to computing the proximate rank. We formally prove this observation below.

 # 6 Computational complexity of proximate rank

Remark 5.3 alludes to an essential difficulty in computing the proximate rank: grouping units with similar (up to sign) incoming weight and bias pairs for merging. The following abstract decision problem, Problem UPC, captures the related task of clustering points in the plane into groups with a fixed maximum uniform radius.[7]

Given $h$ *source points* $x_1, \ldots, x_h \in \mathbb{R}^2$, define an $(r, \varepsilon)$-*cover*, a collection of $r$ *covering points* $y_1, \ldots, y_r \in \mathbb{R}^2$ such that the uniform distance between each source point and its nearest covering point is at most $\varepsilon$ (that is, $\forall i \in \{1, \ldots, h\}, \exists j \in \{1, \ldots, r\}, \|x_i - y_j\|_\infty \leq \varepsilon$).

**Problem UPC.** Uniform point cover, or UPC, is a decision problem. The instances are tuples of the form $(h, r, \varepsilon, X)$ where $h, r \in \mathbb{N}; \varepsilon \in \mathbb{R}^+$; and $X$ is a list of $h$ source points in $\mathbb{R}^2$. The affirmative instances are all tuples $(h, r, \varepsilon, X)$ for which there exists an $(r, \varepsilon)$-cover of the $h$ points in $X$.

**Theorem 6.1.** *Problem UPC is $\mathcal{NP}$-complete.*

*Proof sketch* (Full proof in Appendix C). The main task is to show that UPC is $\mathcal{NP}$-hard ($\forall X \in \mathcal{NP}$, $X \to \text{UPC}$). Since reducibility is transitive, it suffices to give a reduction from the well-known $\mathcal{NP}$-complete problem Boolean satisfiability (Cook, 1971; Levin, 1973). Actually, to simplify the proof, we consider an $\mathcal{NP}$-complete variant of Boolean satisfiability, restricted to formulas with (i) two or three literals per clause, (ii) one negative occurrence and one or two positive occurrences per literal, and (iii) a planar bipartite clause–variable incidence graph.

From such a formula we must construct a UPC instance, affirmative if and only if the formula is satisfiable. Due to the restrictions, the bipartite clause–variable is planar with maximum degree 3, and can be embedded onto an integer grid (Valiant, 1981, §IV). We divide the embedded graph into unit-width tiles of finitely many types, and we replace each tile with an arrangement of source points based on its type. The aggregate collection of source points mirrors the structure of the original formula. The variable tile arrangements can be covered essentially in either of two ways, corresponding to "true" and "false" in a satisfying assignment. The edge tile arrangements transfer these assignments to the clause tiles, where the cover can only be completed if all clauses have at least one true positive literal or false negative literal. Figure 1 shows one example of this construction. ◇

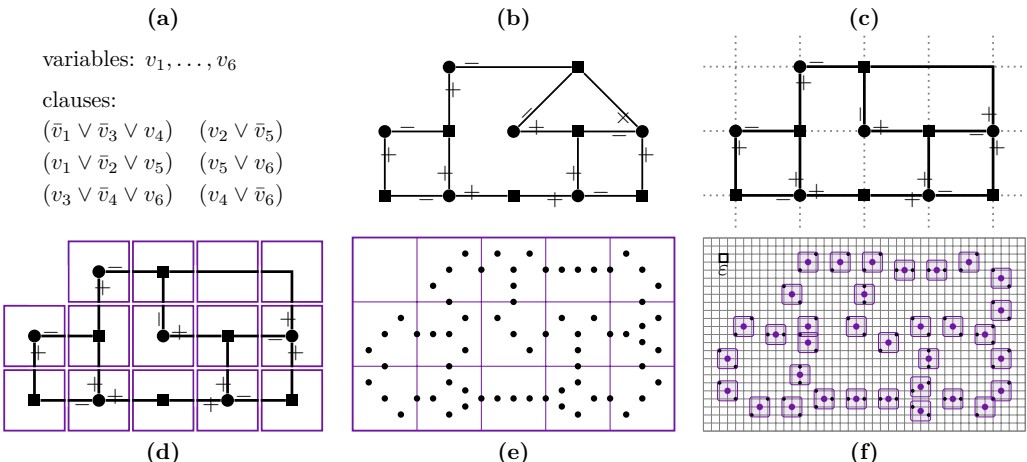

Figure 1: Example of reduction from restricted Boolean satisfiability to Problem UPC. **(a)** A satisfiable restricted Boolean formula. **(b)** The formula's planar bipartite variable–clause invidence graph (circles: variables, squares: clauses, edges: $\pm$ literals). **(c)** The graph embedded onto an integer grid. **(d)** The embedding divided into unit tiles of various types. **(e)** The $h = 68$ source points aggregated from each of the tiles. **(f)** Existence of a $(34, 1/8)$-cover of the source points (coloured points are covering points, with uniform neighbourhoods of radius $1/8$ shown). General case in Appendix C.

---

[7]Problem UPC is reminiscent of known hard clustering problems such as planar $k$-means (Mahajan et al., 2012) and vertex $k$-center (Hakimi, 1964; Kariv and Hakimi, 1979). Supowit (1981, §4.3.2) showed that a Euclidean-distance version is $\mathcal{NP}$-complete. Problem UPC is also related to clique partition on unit disk graphs, which is $\mathcal{NP}$-complete (Cerioli et al., 2004, 2011). We discuss these and other relations in Appendix B.

252 The following decision problem formalises the task of bounding the proximate rank, or equivalently,
253 detecting nearby low-rank parameters. It is $\mathcal{NP}$-complete by reduction from Problem UPC.

254 **Problem PR.** Bounding proximate rank, or PR, is a decision problem. Each instance comprises a
255 number of hidden units $h \in \mathbb{N}$, a parameter $w \in \mathcal{W}_h$, a uniform radius $\varepsilon \in \mathbb{R}^+$, and a maximum
256 rank $r \in \mathbb{N}$. The affirmative instances are those instances where $\mathrm{prank}_\varepsilon(w) \leq r$.

257 **Theorem 6.2.** *Problem PR is $\mathcal{NP}$-complete.*

258 *Proof.* Since UPC is $\mathcal{NP}$-complete (Theorem 6.1), it suffices to show UPC $\rightarrow$ PR and PR $\in \mathcal{NP}$.

259 (UPC $\rightarrow$ PR, the reduction): Given an instance of Problem UPC, allocate one hidden unit per source
260 point, and construct a parameter using the source point coordinates as incoming weights and biases.
261 Actually, to avoid issues with zeros and signs, first translate the source points well into the positive
262 quadrant. Likewise, set the outgoing weights to a positive value. Figure 2 gives an example.

263 Formally, let $h, r \in \mathbb{N}$, $\varepsilon \in \mathbb{R}^+$, and $x_1, \ldots, x_h \in \mathbb{R}^2$. In linear time construct a PR instance with $h$
264 hidden units, uniform radius $\varepsilon$, maximum rank $r$, and parameter $w \in \mathcal{W}_h$ as follows.

1. Define $x_{\min} = \left(\min_{i=1}^h x_{i,1}, \min_{i=1}^h x_{i,2}\right) \in \mathbb{R}^2$, containing the minimum first and second
   coordinates among all source points (minimising over each dimension independently).

2. Define a translation $T : \mathbb{R}^2 \rightarrow \mathbb{R}^2$ such that $T(x) = x - x_{\min} + (2\varepsilon, 2\varepsilon)$.

3. Translate the source points $x_1, \ldots, x_h$ to $x'_1, \ldots, x'_h$ where $x'_i = T(x_i)$. Note (for later)
   that all components of the translated source points are at least $2\varepsilon$ by step (1).

4. Construct the neural network parameter $w = (2\varepsilon, x'_{1,1}, x'_{1,2}, \ldots, 2\varepsilon, x'_{h,1}, x'_{h,2}, 0) \in \mathcal{W}_h$.
   In other words, for $i = 1, \ldots, h$, set $a_i = 2\varepsilon$, $b_i = x'_{i,1}$, and $c_i = x'_{i,2}$; and set $d = 0$.

272 (UPC $\rightarrow$ PR, equivalence): It remains to show that the constructed instance of PR is affirmative if
273 and only if the given instance of UPC is affirmative, that is, there exists an $(r, \varepsilon)$-cover of the source
274 points if and only if the constructed parameter has $\mathrm{prank}_\varepsilon(w) \leq r$.

275 ($\Rightarrow$): If there is a small cover of the source points, then the hidden units can be perturbed so that
276 they match up with the (translated) covering points. Since there are few covering points, many units
277 can now be merged, so the original parameter has low proximate rank.

278 Formally, suppose there exists an $(r, \varepsilon)$-cover $y_1, \ldots, y_r$. Define $\rho : \{1, \ldots, h\} \rightarrow \{1, \ldots, r\}$ such
279 that the nearest covering point to each source point $x_i$ is $y_{\rho(i)}$ (breaking ties arbitrarily). Then for
280 $j = 1, \ldots, r$, define $y'_j = T(y_j)$ where $T$ is the translation defined in step (2) of the construction.
281 Finally, define a parameter $w^\star = (2\varepsilon, y'_{\rho(1),1}, y'_{\rho(1),2}, \ldots, 2\varepsilon, y'_{\rho(h),1}, y'_{\rho(h),2}, 0) \in \mathcal{W}_h$ (in other
282 words, for $i = 1, \ldots, h$, $a_i^\star = 2\varepsilon$, $b_i^\star = y'_{\rho(i),1}$, and $c_i^\star = y'_{\rho(i),2}$; and $d^\star = 0$).

283 Then $\mathrm{rank}(w^\star) \leq r$, since there are at most $r$ distinct incoming weight and bias pairs (namely
284 $y'_1, \ldots, y'_r$). Moreover, $\|w - w^\star\|_\infty \leq \varepsilon$, since both parameters have the same output bias and
285 outgoing weights, and, by the defining property of the cover, for $i = 1, \ldots, h$,

$$\|(b_i, c_i) - (b_i^\star, c_i^\star)\|_\infty = \left\|x'_i - y'_{\rho(i)}\right\|_\infty = \left\|T(x_i) - T(y_{\rho(i)})\right\|_\infty = \left\|x_i - y_{\rho(i)}\right\|_\infty \leq \varepsilon.$$

286 Therefore $\mathrm{prank}_\varepsilon(w) \leq \mathrm{rank}(w^\star) \leq r$.

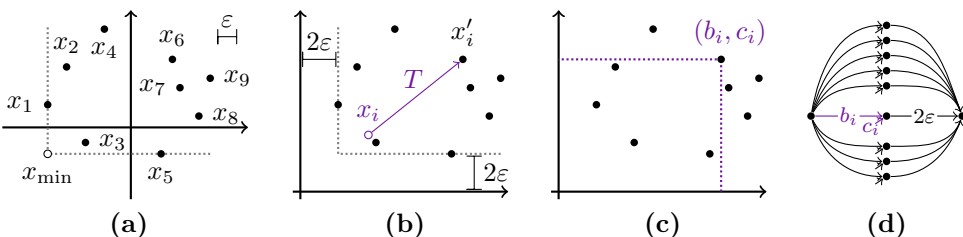

Figure 2: Illustrative example of the parameter construction. **(a)** A set of source points $x_1, \ldots, x_9$.
**(b)** Transformation $T$ translates all points into the positive quadrant by a margin of $2\varepsilon$. **(c,d)** The
coordinates of the transformed points become the incoming weights and biases of the parameter.

($\Leftarrow$): Conversely, since all of the weights and biases are at least $2\varepsilon$, any nearby low-rank parameter implies the approximate mergeability of some units. Therefore, if the parameter has low proximate rank, there is a small cover of the translated points, and, in turn, of the original points.

Formally, suppose $\mathrm{prank}_\varepsilon(w) \leq r$, with $w^\star \in \bar{B}_\infty(w; \varepsilon)$ such that $\mathrm{rank}(w^\star) = r^\star \leq r$. In general, the only ways that $w^\star$ could have reduced rank compared to $w$ are the following (cf. Algorithm 4.1):

1. Some incoming weight $b_i$ could be perturbed to zero, allowing its unit to be eliminated.

2. Two units $i, j$ with $(b_i, c_i)$ and $(b_j, c_j)$ within $2\varepsilon$ could be perturbed to have identical incoming weight and bias, allowing them to be merged.

3. Two units $i, j$ with $(b_i, c_i)$ and $-(b_j, c_j)$ within $2\varepsilon$ could be perturbed to have identically negative weight and bias, again allowing them to be merged.

4. Some group of $m \geq 1$ units, merged through the above options, with total outgoing weight within $m\varepsilon$ of zero, could have their outgoing weights perturbed to make the total zero.

By construction, all $a_i, b_i, c_i \geq 2\varepsilon > 0$, immediately ruling out (1) and (3). Option (4) is also ruled out because any such total outgoing weight is $2m\varepsilon > m\varepsilon$. This leaves option (2) alone responsible. Thus, there are exactly $r^\star$ distinct incoming weight and bias pairs among the units of $w^\star$. Denote these pairs $y'_1, \ldots, y'_{r^\star}$—they constitute an $(r^\star, \varepsilon)$-cover of the incoming weight and bias vectors of $w, x'_1, \ldots, x'_h$ (as $w^\star \in \bar{B}_\infty(w; \varepsilon)$). Finally, invert $T$ to produce an $(r^\star, \varepsilon)$-cover of $x_1, \ldots, x_h$, and add $r - r^\star$ arbitrary covering points to extend this to the desired $(r, \varepsilon)$-cover.

(PR $\in \mathcal{NP}$): We must show that an affirmative instance of PR can be verified in polynomial time, given a certificate. Consider an instance $h, r \in \mathbb{N}$, $\varepsilon \in \mathbb{R}^+$, and $w = (a_1, b_1, c_1, \ldots, a_h, b_h, c_h, d) \in \mathcal{W}_h$. Use as a certificate a partition[8] $\Pi_1, \ldots, \Pi_J$ of $\{ i \in \{1, \ldots, h\} : \mathrm{abs}(b_i) > \varepsilon \}$, such that (1) for each $\Pi_j$, for each $i, k \in \Pi_j$, $\|\mathrm{sign}(b_i) \cdot (b_i, c_i) - \mathrm{sign}(b_k) \cdot (b_k, c_k)\|_\infty \leq 2\varepsilon$; and (2) at most $r$ of the $\Pi_j$ satisfy $\sum_{i \in \Pi_j} \mathrm{sign}(b_i) \cdot a_i > \varepsilon \cdot |\Pi_j|$. The validity of such a certificate can be verified in polynomial time by checking each of these conditions directly.

It remains to show that such a certificate exists if and only if the instance is affirmative. If $\mathrm{prank}_\varepsilon(w) \leq r$, then there exists a parameter $w^\star \in \bar{B}_\infty(w; \varepsilon)$ with $\mathrm{rank}(w^\star) \leq r$. The partition computed from Stage 2 of COMPRESS$(w^\star)$ satisfies the required properties for $w \in \bar{B}_\infty(w^\star; \varepsilon)$.

Conversely, given such a partition, for each $\Pi_j$, define $v_j \in \mathbb{R}^2$ as the centroid of the bounding rectangle of the set of points $\{ \mathrm{sign}(b_i) \cdot (b_i, c_i) : i \in \Pi_j \}$, that is,

$$v_j = \frac{1}{2} \left( \max_{i \in \Pi_j} \mathrm{abs}(b_i) + \min_{i \in \Pi_j} \mathrm{abs}(b_i), \ \max_{i \in \Pi_j} \mathrm{sign}(b_i) \cdot c_i + \min_{i \in \Pi_j} \mathrm{sign}(b_i) \cdot c_i \right).$$

All of the points within these bounding rectangles are at most uniform distance $\varepsilon$ from their centroids. To construct a nearby low-rank parameter, follow the proof of Theorem 5.1 using $\Pi_1, \ldots, \Pi_J$ and $v_1, \ldots, v_J$ in place of their namesakes from Algorithms 5.1 and 5.2. Thus $\mathrm{prank}_\varepsilon(w) \leq r$. $\qquad\square$

# 7 Discussion

In this paper, we have studied losslessly compressible neural network parameters, measuring the size of a network by the number of hidden units. Losslessly compressible parameters comprise a measure zero subset of the parameter space, but this is a rich subset that stretches throughout the entire parameter space (Anonymous, 2023). Moreover, the neighbourhood of this region has nonzero measure and comprises approximately compressible parameters.

It's possible that part of the empirical success of deep learning can be explained by the proximity of learned neural networks to losslessly compressible parameters. Our theoretical and algorithmic contributions, namely the notions of rank and proximate rank and their associated algorithms, serve as a foundation for future research in this direction. In this section, we outline promising next steps for future work and discuss limitations of our approach.

---

[8]It would seem simpler to use a nearby low-rank parameter itself as the certificate, which exists exactly in affirmative cases by definition of the proximate rank. Unfortunately, an arbitrary nearby low-rank parameter is unsuitable because the parameter could have unbounded description length, leading to the certificate not being verifiable in polynomial time. By using instead this partition we essentially establish that in such cases there is always also a nearby low-rank parameter with polynomial description length.

**Limitations of the lossless compressibility framework.** Section 4 offers efficient algorithms for optimal lossless compression and computing the rank of neural network parameters. However, the rank is an idealised notion, serving as a basis for the theory of proximate rank. One would not expect to find compressible parameters in practice, since numerical imprecision is likely to prevent the observation of identically equal, negative, or zero weights in practice. Moreover, the number of units is not the only measure of a network's description length. For example, the sparsity and precision of weights may be relevant axes of parsimony in neural network modelling.

Returning to the deep learning context—there is a gap between lossless compressibility and phenomena of approximate compressibility. In practical applications and empirical investigations, the neural networks in question are only approximately preserved the function, and moreover the degree of approximation may deteriorate for unlikely inputs. Considering the neighbourhoods of losslessly compressible parameters helps bridge this gap, but there are approximately compressible neural networks beyond the proximity of losslessly compressible parameters, which are not accounted for in this approach. More broadly, a comprehensive account of neural network compressibility must consider architectural redundancy as well as redundancy in the parameter.

**Tractable detection of proximity to low-rank parameters.** An important direction for future work is to empirically investigate the proximity of low-rank neural networks to the neural networks that arise during the course of successful deep learning. Unfortunately, our main result (Theorem 6.2) suggests that detecting such proximity is computationally intractable in general, due to the complex structure of the neighbourhoods of low-rank parameters.

There is still hope for empirically investigating the proximate rank of learned networks. Firstly, $\mathcal{NP}$-completeness does not preclude efficient approximation algorithms, and approximations are still useful as a one-sided test of proximity to low-rank parameters. Algorithm 5.1 provides a naive approximation, with room for improvement in future work. Secondly, Theorem 6.2 is a worst-case analysis—Section 6 essentially constructs pathological parameters poised between nearby low-rank regions such that choosing the optimal direction of perturbation involves solving (a hard instance of) Boolean satisfiability. Such instances might be rare in practice (cf. the related problem of $k$-means clustering; Daniely et al., 2012). As an extreme example, detecting proximity to merely compressible parameters ($r = h - 1$) permits a polytime solution based on the reducibility conditions.

**Towards lossless compressibility theory in modern architectures.** We have studied lossless compressibility in the simple, concrete setting of single-hidden-layer hyperbolic tangent networks. Several elements of our approach will be useful for future work on more modern architectures. At the core of our analysis are structural redundancies arising from zero, constant, or proportional units (cf. reducibility conditions (i)–(iii)). In particular, the computational difficulty of bounding the proximate rank is due to the approximate merging embedding a hard clustering problem. These features are not due to the specifics of the hyperbolic tangent, rather they are generic features of any layer in a feed-forward network component.

In more complex architectures there will be additional or similar opportunities for compression. While unit negation symmetries are characteristic of odd nonlinearities, other nonlinearities will exhibit their own affine symmetries which can be handled analogously. Further redundancies will arise from interactions between layers or from specialised computational structures.

## 8 Conclusion

Towards a better understanding of complexity and compressibility in learned neural networks, we have developed a theoretical and algorithmic framework for *lossless* compressibility in single-hidden-layer hyperbolic tangent networks. The *rank* is a measure of a parameter's lossless compressibility. Section 4 offers efficient algorithms for performing optimal lossless compression and computing the rank. The *proximate rank* is a measure of proximity to low-rank parameters. Section 5 offers an efficient algorithm for approximately bounding the proximate rank. In Section 6, we show that optimally bounding the proximate rank, or, equivalently, detecting proximity to low-rank parameters, is $\mathcal{NP}$-complete, by reduction from Boolean satisfiability via a novel hard clustering problem. These results underscore the complexity of losslessly compressible regions of the parameter space and lay a foundation for future theoretical and empirical work on detecting losslessly compressibile parameters arising while learning with more complex architectures.

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
