# OpenReview forum: "Computational Complexity of Detecting Proximity to Losslessly Compressible Neural Network Parameters"
_NeurIPS.cc/2023/Conference — Submitted to NeurIPS 2023_

### Official Review · Reviewer_aLjx · 2023-06-30

**Soundness:** 3 good
**Presentation:** 4 excellent
**Contribution:** 3 good
**Rating:** 4
**Confidence:** 2

**Summary:**

This paper studies the replication of the same function using a smaller network. It provides a procedure for achieving optimal lossless compression in the context of single-hidden-layer hyperbolic tangent networks.

This paper introduces the idea of 'rank' of a parameter, defined as the least number of hidden units needed to replicate the same function. The paper further defines the 'proximate rank' of a parameter as the rank of the most compressible parameter within a limited L-inf neighborhood.

The paper also demonstrates that estimating the proximate rank is an NP-complete problem through a reduction from Boolean satisfiability using a geometric problem that involves covering points in a plane with small squares.

**Strengths:**

1. This is a theoretical treatment of a very important problem and the results are interesting.

2. The paper shows that the problem of bounding the approximate rank below a given value is an NP-complete decision problem.

3. Paper is well presented and clear.

**Weaknesses:**

1. The results do not seem to be important. Specifically, does anyone care if instead of totally lossless compression, we have an infinitesimal error in representation?

2. The algorithm given in this paper is only applicable to single-hidden layer hyperbolic tangent networks, without obvious ways of extending it to more general cases.

3. The paper aims to provide a theoretical ground for network compression, but did not argue extensively on its connection on the empirical success of existing network compression literature.

**Questions:**

1. Is it possible to define a  more relaxed version of "rank", that is more applicable to the existing neural network compression methods?

2. There has been some theoretical treatments of the problem of network compression. For instance, https://arxiv.org/abs/2206.05604 approach the problem with linear approximation. What are the connections to your work?

**Limitations:**

1. In practice it is allowed that the compressed networks have slight performance drop compared to the original network, but this paper does not consider this problem.

2. The proposed algorithm only considers the single-hidden layer hyperbolic tangent networks, which is not often employed in practice.

---

> ### Author Rebuttal · Authors · 2023-08-10
>
> We thank reviewer aLjx for their thorough review of our work. We thank the reviewer for pointing out strengths in our paper including our interesting results and clear presentation. We also thank the reviewer for pointing out several legitimate limitations in the scope of our analysis including some that we had not sufficiently acknowledged.
>
> We address each of the reviewer's listed weaknesses, questions, and limitations as follows. We can offer limited revisions of the paper itself at this stage. However, we hope that the reviewer might reconsider their recommendation to reject the paper based on the potential for these concerns to be addressed with future work, with our paper evaluated as a strong 'first step' in this new research direction (in line with the reviewer's 'good' and 'excellent' ratings, and listed strengths).
>
> **Weakness 3, questions 1 & 2:** We also appreciate the reviewer taking the time to point out what we agree is a legitimate weakness of our submission that we have not previously acknowledged: we fail to establish a connection between our framework and existing theory on neural network compression. The best response we can offer is as follows:
> If accepted, we plan to acknowledge this potential connection in the discussion section as an important direction for future work. However, we must leave the exploration to future work because, having come to this problem from the study of neural network training dynamics and functional equivalence (rather than compression), we are not sufficiently familiar with the existing literature to provide any concrete results yet. It does make sense to us that compression-based generalisation bounds might be achievable for networks with low proximate rank based on their proximity to simpler, losslessly compressible parameters.
>
> **Weakness 1:** More broadly, we offer an alternative motivation for caring about networks being within epsilon of a losslessly compressible network. We expect the dynamics and statistics of learning with parameters that are *at all* close to losslessly compressible parameters will be influenced by the structure of the degenerate critical points located at those losslessly compressible parameters. We note that epsilon could reasonably be much larger than machine precision (rather than infinitesimal). We refer the reviewer to the citations on manuscript lines 29 to 29 on these topics, and are happy to discuss further if required.
>
> **Weakness 2 & limitation 2:** We agree that a major limitation to the scope of our study is the choice of architecture, with single-hidden-layer networks being of limited use in modern deep learning practice. However, we view this architecture as an ideal 'first step' since it has simple enough structure to fully characterise and present in one paper, and also serves as a meaningful basis for future work to build upon. We refer the reviewer to the top-level author rebuttal where we expand on this point in more detail.
>
> **Limitation 1:** The reviewer has pointed to a gap between our results and practical applications of network compression, which amounts to the difference between lossless and lossy compression. We put it this way:
>
> * We study *exactly* preserving the *function* implemented by the network on *all inputs*; whereas
> * in practice it may suffice to *approximately* preserve the *loss* achieved by the network function on *a sample of inputs* (such as the training or validation data).
>
> We agree that this is an important gap. We are excited about future work bridging this gap. However, drawing on the above-mentioned motivations for studying losslessly compressible parameters and their neighbourhoods (for their implications on training dynamics and statistics of neural networks), we think that there is also a case to be made for considering the lossless compression setting. Moreover, one could consider compromise settings such as:
>
> * studying *approximately* preserving the function implemented by the network on *all inputs*, or
> * studying *exactly* preserving the function implemented by the network on *a sample of inputs*.
>
> We are excited about future work addressing each of these directions as well.

---

> > ### Comment · Reviewer_aLjx · 2023-08-10
> > **I am keeping my score**
> >
> > Thanks for your work and explanation.  I am glad that you agree with most of my original comments.  I am keeping my score and looking forward to seeing the improved version of the paper in another venue

---

### Official Review · Reviewer_Coti · 2023-07-04

**Soundness:** 3 good
**Presentation:** 4 excellent
**Contribution:** 3 good
**Rating:** 7
**Confidence:** 2

**Summary:**

The paper studies the idealized phenomenon of lossless compressibility, whereby an identical
function can be implemented with a smaller network, in the setting of single-hidden-layer hyperbolic tangent networks.  It introduces the notion of rank as the minimal number of hidden units (used to measure the "size" of a network) required to implement a given NN function and a constructive algorithm for computing it, as well as the notion of *proximate rank* as the rank of the most compressible parameter vector within a small L_infty norm ball and an algorithm for upper-bounding it. The paper further shows that bounding the proximate rank of a given parameter vector is an NP-complete problem.


**Strengths:**

While I'm not an expert on algorithms and cannot comment on correctness or the novelty of the approach,
I followed the high-level argument and believe the contribution is significant. The paper studies a fundamental aspect of deep learning, i.e., the compressibility and description length of neural networks, which is mostly dominated by empirical research and demands better theoretical understanding. The paper goes beyond the classical setting of lossless compressibility (c.f. Sussmann 1992) by introducing the notion of proximate rank and proving a basic hardness results for it, which will hopefully lay a foundation for future studies of this topic.

**Weaknesses:**

Section 6 (Computational complexity of proximate rank) is a bit hard to follow for non-experts. One possibility could be to include a discussion at the beginning of Section 6 that summarizes the proof at a high-level.

**Questions:**

1. I'm curious how the authors define "description length" ("polynomial description length" in footnote 8). It appears there's a widely accepted notion of description length in this line of work?

2. Related to the above, could the authors comment on how their notion of NN size (minimum number of hidden units) relates to alternative description length such as Kolgomorov complexity?


**Limitations:**

The authors adequately addressed the limitations.

---

> ### Author Rebuttal · Authors · 2023-08-10
>
> We thank reviewer Coti for their review of our work. We thank the reviewer for acknowledging various strengths of our work including its clear presentation, and we graciously thank them for the recommendation that our paper be accepted.
>
> **Weakness:** The main weakness listed by the reviewer is an issue of presentation. We are appreciative of the reviewer's suggestion. Upon revisiting the manuscript we have also thought that the structure of section 6, which (1) begins with the abstract clustering problem and then (2) returns to the neural network problem, is perhaps suboptimal for the ML audience. We are considering revamping section 6 so as to incorporate some or all of the following cosmetic changes:
>
> 1. Incorporate the suggestion of the reviewer to begin with a high-level conceptual overview of the proof (through which we are optimistic we can indeed achieve greater clarity for non-specialist readers).
> 2. Changing the order of the proof sketches so that the section starts with the formulation of problem PR rather than the abstract clustering problem.
> 3. Rewriting the proof sketch for the clustering problem to make clearer the meaning of the construction in terms of neural network weights and biases, rather than just abstract points in the plane (through which we hope that our ML audience will be more readily able to follow both this component of the proof and also the whole proof of the main result for neural networks).
> 4. possibly deferring some additional details of the abstract problem's hardness proof sketch to the appendix, rather than keeping them in the main paper (since they are not crucially important to the 'story' of the main paper).
>
> We welcome feedback from our reviewers on these presentation proposals.
>
> **Questions:** The reviewer also asked two questions which we are happy to briefly answer as follows.
>
> 1. **Description length (of footnote 8):** Yes. This answer are somewhat technical and require delving into the details of definitions of complexity classes involving polynomial-time Turing machines, which we have mostly kept out of the paper, except for in a few places like footnote 8. In this case, by 'polynomial description length' we refer to the length of the verifier's 'certificate', measured in Turing Machine tape symbols used for encoding the certificate. This length must be polynomial as a function of the length of the encoded problem instance in tape symbols. The problem instance will include a neural network parameter, whose components are, by convention, going to be encoded in some fixed-precision numerical representation. For a nearby parameter to act as a polynomial-time verifiable certificate, it must also be possible to encode the nearby parameter using a similar level of precision, otherwise the certificate encoding size won't be polynomial in the instance encoding size. To guarantee a polynomial-size certificiate we therefore use a certificate that does not involve a real parameter vector but rather a discrete partition of the set of hidden units of the network (represented as integers).
>
> 2. **Rank and description length:** We have not thought extensively about this connection. Consider Kolmogorov complexity of a network defined as the length of the shortest program for some fixed universal Turing machine (ignore the dependence of Kolmogorov complexity on the choice of UTM) that outputs a fixed-width decimal representation of some network's weigh vector.
>     * This might not be so closely related to rank, since, for example, small programs can, using loops, output parameters for networks with many units if the units have similar structure. For example, consider a network with 100 units that have incoming weights i=1,2,3,4,...,100. A simple program can output this irreducible network, but a complex program may be required to precisely specify a 10-unit network with highly-precise decimal weights. With these differences in mind, we would argue that the number of units is itself a useful measure of complexity tailored to neural networks, and may be more appropriate than some kind of Kolmogorov complexity.
>     * On the other hand, lossless compressibility is surely important for computing a minimum Kolmogorov description length of a neural network *function*. One would not want to waste any bits representing units redundantly if one is planning to use a neural network parameter as a means to describe a particular function. Thus one would probably want to encode the compressed version of the parameter rather than any other version.
>
> We would be happy to expand on these questions in the discussion period.
>
> (Edit: Minor edit slightly after rebuttal deadline to expand on answer to question 2.)

---

> > ### Comment · Reviewer_Coti · 2023-08-15
> >
> > Thanks for the insightful response. I'm maintaining my score at the moment.
> > The revision plan looks good to me, and I think points 1 and 3 would be especially helpful.

---

### Official Review · Reviewer_t3vj · 2023-07-06

**Soundness:** 3 good
**Presentation:** 4 excellent
**Contribution:** 2 fair
**Rating:** 6
**Confidence:** 3

**Summary:**

The authors propose two notions for studying neural network complexity with, on single hidden layer neural networks. The first is _rank_, which corresponds to the smallest number of parameters that can produce a network that is functionally equivalent to the original, and the second is _proximate rank_, which is the rank of the network with smallest rank in an $L^\inf$ neighborhood of the original model('s parameters). The authors propose an algorithm for computing rank, and propose a heuristic algorithm for upper bounding proximate rank. They also show that exactly computing proximate rank is NP-complete, so any future study will have to rely on approximations / bounds on this quantity.

**Strengths:**

- The notions the authors introduce are interesting and future work can indeed utilize them in interesting ways. The metrics they propose importantly diverge from metrics such as $\ell_p$-compressibility in that they are defined based on functional equivalence, which might imply them being more useful for future learning theoretic research.
- The paper is very well structured, and the exposition is clear and easy to follow. Connections to previous literature are made clear (but see below).

**Weaknesses:**

The paper is beset by two important problems:
- As the authors widely acknowledge, their notions are defined on single layer neural networks with tanh activation, with no clear map for generalization to more realistic setups. It would be valuable to see a roadmap for this, as it would help evaluate whether proximate rank is a potentially useful theoretical notion for future research or in the worst case a "dead end" in terms of applicability to realistic ML models.
- The authors do not discuss how or why their notions of rank and proximate rank could be useful for future research on generalization in deep learning (or robustness etc.). Given the venue the paper is submitted, I am assuming that the authors believe that these results might have learning theoretical implications down the line, but they are mostly silent on this issue. They correctly observe that approximate compressibility research has been utilized for such aims, but do not opine on how their alternative notions should be superior or even have qualitatively different contributions to such research. While this is not a problem per se in general, it is more so for a machine learning conference.

**Questions:**

- L68: Is there a common theoretical / practical reason for previous literature having given little consideration to cases other than "minimal networks"?
- L75: What are canonical equivalent parameters?
- I think a broader discussion of the relation of current work to "compressibility" research would be helpful for the reader, especially so for the generalization-related subset of this research, e.g. [1] being one of the most well-known.

[1] Arora, S., Ge, R., Neyshabur, B., & Zhang, Y. (2018). Stronger Generalization Bounds for Deep Nets via a Compression Approach. ICLR.

**Limitations:**

The authors are transparent about the limitations of their work.

---

> ### Author Rebuttal · Authors · 2023-08-10
>
> We thank reviewer t3vj for their thorough review of our submission. We thank the reviewer for pointing out several strengths in our paper including the potential of our results and our clear structure and exposition. We especially thank the reviewer for their recommendation that the paper be (borderline) accepted.
>
> In the weaknesses section of the review, the reviewer has raised two important problems arising from the paper's limited scope. We think these concerns are legitimate and we aim to comment on them below. Unfortunately, we are not able to offer substantial revisions to the manuscript to completely address these limitations in this submission. Nevertheless, we invite the reviewer to reconsider their 'borderline' recommendation for acceptance, based on an evaluation of our submission as a standalone first step in an important new research direction, leaving the concerns to be comprehensively addressed in future work.
>
> **Problem 1:** As noted we are cognizant of the limited direct applicability of the single-hidden-layer tanh architecture. We appreciate the reviewer highlighting the importance of this issue for their evaluation of the work's potential importance. We refer the reviewer to our top-level author rebuttal where we have attempted to sketch a roadmap for generalisation of our results to a more general setting. In summary, we still consider the single-hidden-layer tanh case to be a meaningful setting to study since within this setting arise some sources of reducibility / compressibility that we expect to be fundamental to general feed-forward architectures.
>
> **Problem 2 (also question 3):** We also appreciate the reviewer highlighting the importance of connecting our work to existing work in deep learning theory. If accepted, we plan to add to the discussion section a proposal for future work to explore the connection between our results and compressibility-based generalisation theory. It does make sense to us that compressibility-based generalisation bounds might be achievable for networks with low proximate rank based on their similarity to simpler, losslessly compressible parameters. However, we must leave the exploration to future work because, having come to this problem from the study of neural network training dynamics and functional equivalence (rather than compression), we are not sufficiently familiar with the existing literature to provide any concrete results yet. Because of our lack of detailed background in this area we are especially grateful to the reviewer for their valuable literature recommendation, and would highly value any further recommendations if the reviewer has them to hand.
>
> ---
>
> The author also asked two other questions (question 3 addressed above) which we are happy to answer.
>
> **Question 1:** We thank the reviewer for this great question about the prior neglect of non-minimal networks in theoretical work on functional equivalence. Based on our survey of the stated justifications given in prior work, these fall into two camps:
>
> * Some prior work has explicitly cited the negligible volume of sets of nonminimal parameters as a rationale for excluding them from analysis. This is based on a practical intuition that such networks have zero probability of occurring at random, and therefore they are unlikely to be relevant for learning.
>
>     We dispute this point since relevant networks are trained based on data, not randomly generated. The comments in the manuscript on lines 29 to 39 capture our justification for why we think these parameters are nevertheless interesting to study, and we are happy to elaborate further on this topic on request.
>
> * Most prior work acknowledges the assumptions made as temporary, and they leave discussing the general case to future work.
>
>     We can hazard a guess that such work is motivated by theoretical elegance of achieving a 'neat' *almost*-identifiability result. For example, one *wants to* say that a neural network function determines it's parameter. One can't say *that*, but, one *can* say that a function determines its parameter up to 'simple and uninteresting symmetries' such as unit permutations (which are anyway an artefact of implementation---why do we need to order units in a hidden layer, really? one might say).
>
>     We would say that our main conceptual innovation in the supplementary preprint (Anonymous, 2023) is to use an algorithmic approach to achieve a *relatively* 'neat' characterisation of the fully general case including these parameters. The resulting insights into functional equivalence are the basis for our approach to studying (approximate) lossless compressibility in the present work.
>
> **Question 2:** Canonical equivalent parameters: Of all the parameters that are functionally equivalent to a given parameter, pick one to be a 'canonical' (=standard) representation of that function. The supplementary preprint cited develops an algorithm for finding such parameters given any input parameter (such that functionally equivalent parameters will always be converted to the same canonical equivalent parameter). The broader point being made here is that the algorithm in the attached preprint implements lossless compression as part of the process of standardising/canonicalising the parametric representation of a given neural network function, but the algorithm they implement is not optimised for lossless compression itself.
> It is apparent that this phrasing is not immediately clear as we mistakenly thought---we will work to revise the text so that the meaning is more understandable.
>
> ---
>
> A minor technical correction: The reviewer's summary states that we show "exactly computing proximate rank is NP-*complete*". Actually, it is NP-*hard*, as a non-decision problem---the decision problem, *bounding* proximate rank, is NP-complete. This is a common slip and we have endeavoured to be precise in our terminology. If the reviewer noted anywhere in our manuscript where we made this mistake, please advise!

---

> > ### Comment · Reviewer_t3vj · 2023-08-15
> > **Thanks for the comments**
> >
> > I thank the authors for their comments, and appreciate the further discussion re. limitations and potential of their work, as well as the correction re. Theorem 6.1. I think the additional discussion will help the conference audience in engaging with their findings. Although the limitations discussed above prevent me from doing so drastically, I raise my score to reflect their additional contributions.
> >
> > Below are a number of (non-exhaustive) additional relevant references:
> >
> > [1] T. Suzuki, H. Abe, and T. Nishimura, “Compression Based Bound for Non-compressed Network: Unified Generalization Error Analysis of Large Compressible Deep Neural Network,” presented at the International Conference on Learning Representations, 2020.
> > [2] T. Suzuki et al., “Spectral Pruning: Compressing Deep Neural Networks via Spectral Analysis and its Generalization Error,” in Proceedings of the Twenty-Ninth International Joint Conference on Artificial Intelligence, Yokohama, Japan: International Joint Conferences on Artificial Intelligence Organization, Jul. 2020, pp. 2839–2846.
> > [3] M. Barsbey, M. Sefidgaran, M. A. Erdogdu, G. Richard, and U. Simsekli, “Heavy Tails in SGD and Compressibility of Overparametrized Neural Networks,” in Advances in Neural Information Processing Systems, Curran Associates, Inc., 2021, pp. 29364–29378.
> > [4] D. Hsu, Z. Ji, M. Telgarsky, and L. Wang, “Generalization Bounds via Distillation.” arXiv.2104.05641, Apr. 12, 2021.
> > [5] M. Sefidgaran, A. Gohari, G. Richard, and U. Simsekli, “Rate-Distortion Theoretic Generalization Bounds for Stochastic Learning Algorithms,” in Proceedings of Thirty Fifth Conference on Learning Theory, PMLR, Jun. 2022, pp. 4416–4463.

---

> > > ### Author Response · Authors · 2023-08-21
> > > **Thank you**
> > >
> > > Thank you kindly for these additional valuable literature recommendations and for reconsidering your recommendation.

---

### Official Review · Reviewer_ubUt · 2023-07-08

**Soundness:** 3 good
**Presentation:** 3 good
**Contribution:** 3 good
**Rating:** 6
**Confidence:** 1

**Summary:**

The paper deep-dives into the concept of lossless network compressibility. It presents an algorithm for optimal lossless compression in single-hidden-layer hyperbolic tangent networks, which can in part generalize to other more relevant feedforward architectures. The authors introduce the novel concept of "proximate rank", which measures network complexity, and demonstrate that bounding the proximate rank is an NP-complete problem, thereby suggesting that the problem of finding highly compressible networks is very hard.

**Strengths:**

- originality: The paper introduces novel approaches to obtain lossless compression and introduces the novel concept of proximate rank as a measure of neural network complexity. Furthermore the paper also introduces novel techniques to prove that bounding the proximate rank measure is an NP-complete problem. In summary, many novel contributions which, as the authors argue, lay the foundations for future work identifying losslessly compressible parameters in deep learning structures.

- quality: The quality of the paper seems sound. The topic is a bit away from my area of expertise and I did not check the math in detail though.

- clarity: the paper is very well written.

- significance: Lossless compressibility is a fundamental topic in deep learning, particularly since most recent state-of-the-art research involve very large foundation models. Having a theoretical framework and some robust approaches to measure network complexity and compressibility are key to advance the field of deep learning and the applicability of modern architectures.

**Weaknesses:**

Single-hidden-layer hyperbolic tangent network can be a good subject to start developing a theoretical framework and a set of tools to measure and optimize network complexity and compressibility, but as the authors point out, this architecture is of little relevance otherwise (for current research). This issue, which is raised by the authors, is a minor weakness but it does limit the potential impact of the present work.

**Questions:**

No questions for the authors.

**Limitations:**

Authors have addressed all the limitations clearly.

---

> ### Author Rebuttal · Authors · 2023-08-10
>
> We thank reviewer ubUt for taking the time to review our paper, especially given it is outside of their specialisation.
>
> The main weakness identified by the reviewer (and some other reviewers) is the limitation in the technical scope of the study (to single-hidden-layer hyperbolic tangent networks). The reviewer points out that this scope limits the potential impact of the work. We agree that this is an important limitation but we insist that our results are themselves a meaningful first step towards a more general analysis in more complex architectures, since the compressibility structure present in a single hidden layer is also inherited by a multi-layer architecture (alongside other additional structure), for example. We refer the reviewer to the top-level author rebuttal where we lay out this case in more detail.
>
> We thank the reviewer for their kind comments on the clarity of the paper. We are of course interested in improving the accessibility of the paper as much as possible (acknowledging that it also involves a fundamentally complex technical contribution). We would be appreciative if the reviewer noted any particular areas that were difficult to follow for non-specialists, but could have been made more clear.
>
> Either way, we thank the reviewer again for the attention they have given to our work.

---

### Author Rebuttal · Authors · 2023-08-10

We thank all of our reviewers for their time spent thoroughly reviewing our submission. As a supplement to our individual review responses, we want to take the opportunity here in the top-level author rebuttal to expand on a common thread raised in three of our four reviews, namely the limitation of the technical scope of the paper to the setting of single-hidden-layer hyperbolic tangent networks. As pointed out by reviewer t3vj, the extent to which our results can be generalised to more modern architectures is crucial to determining if our framework is a *'potentially useful theoretical notion for future research or in the worst case a "dead end" in terms of applicability to realistic ML models.'*

Our contention is twofold:

1. firstly that **our results can potentially be generalised to more modern architectures**, with future work; and
2. secondly that **the contribution of this paper is a meaningful first step in that direction,** not *merely* a toy example.

To defend this contention, please allow us to reiterate and expand upon the comments made in the discussion section 'Towards lossless compressibility theory in modern architectures' (which, of course, we will make a point to expand in the manuscript if accepted, since this is obviously an important element of the discussion that could have been more detailed in the paper itself):

* In more modern architectures such as feed-forward networks with many layers and with different activation functions (e.g. ReLU), most of the sources of redundancy underpinning our analysis remain present. In particular, it is still possible to merge units that implement proportional functions, or eliminate units that have zero incoming or outgoing weights.
    * Note that the merging of proportional units is the ultimate source of the main hardness result, because identifying the optimal merging strategy is shown to embed a hard clustering problem. In other words, we expect that this result will generalise to more complex architectures.
    * The fourth reducibility condition, arising from the odd property of tanh, is not generic to other activation functions, but neither is it very fundamental to our results.
* In other architectures or with different activation functions, there will be additional sources of redundancy that we can work to characterise (for lossless compressibility) and efficiently remove (for lossless compression).
    * In the case of ReLU activations, the well-known positive scaling symmetry is at play, which can be straightforwardly combined with the mergeability conditions to identify additional pairs of mergeable units.
    * In the case of multi-layer architectures, the possibility for redundancy to arise *between* layers enters the picture.
        * A simple but extreme example would be that if any layer implements the zero function, then the whole network can be compressed away.
        * Another example would be if one ReLU layer implements the identity function, then this layer commutes with adjacent layers, and can be removed.
        * Further examples of more subtle interactions between layers clearly exist but are yet to be systematically catalogued---to the best of our knowledge this kind of inter-layer redundancy has not been previously studied (though some works on affine symmetries in multi-layer ReLU networks have identified the potential by way of excluding these parameters from study).
    * Further symmetries will arise in the presence of residual connections, or from modern computational elements such as attention heads / transformer blocks.
* However, even in the presence of all of these additional sources of redundancy, the simple within-layer redundancies mentioned above (from merging and eliminating units) will still be present as well. **Thus our analysis is already partially relevant for future work generalising our results to deeper architectures.**

Overall, we consider shallow tanh networks as a simple architecture in which arises a key part of the theory of lossless compressibility arising in any feed-forward layer of any architecture (plus at this stage already computational complexity of the relaxed version arises and can be studied). This makes it an ideal setting in which to begin conducting our study of lossless compressibility.

---

> ### Comment · Reviewer_Coti · 2023-08-14
>
> Thanks for the thoughtful response. One clarifying question that I think many reviewers would appreciate is this:
> since the odd function property of tanh is not very fundamental to the results, do you expect the results to continue to hold if we replace tanh with Relu (perhaps after some minor adjustments to the existing proofs)?

---

> > ### Author Response · Authors · 2023-08-21
> > **Clarificiation of the role of odd property and positive scaling symmetry in our main results.**
> >
> > Good question, we are happy to clarify.
> >
> > The odd property of tanh is not fundamental to the results in the following sense. Consider if the odd property were removed, but the activation function otherwise had the same properties as tanh. Then all of the algorithms and proofs in the paper would straightforwardly generalise with minor modifications to the algorithms and proofs:
> >
> > * **Reducibility:** First, reducibility condition (iv) would be removed. The other three reducibility conditions would remain, and for now we further assume that they would continue to be exhaustive.
> > * **Algorithms:** In the lossless compression algorithm, rank algorithm, and proximate rank bounding algorithm, the (approximate) partition of the units by the absolute incoming weight and bias vector could be replaced with an (approximate) partition by the units by the signed incoming weight and bias instead.
> > * **Hardness:** For the hardness proof, the result is unchanged, and the exact same construction would work, as this construction does not make use of the odd property. In face it may be possible to simplify the construction since the provided construction in part functions to avoid reducibility condition (iv), and this is no longer necessary.
> >
> > Consider now the separate question of how the results could be generalised if the activation function were replaced with ReLU instead of tanh. In addition to lacking the odd property of tanh, ReLU has some further additional properties. For example, ReLU displays the aforementioned positive scaling symmetry, where for $c > 0$, $\mathrm{ReLU}(z) = c \cdot \mathrm{ReLU}(z/c)$. Incorporating this symmetry would lead to further modifications of the reducibility conditions and algorithms.
> >
> > * **Reducibility:** A new fourth reducibility condition could be added whereby if the incoming weight and bias pair of two neurons are related by a positive constant, then these neurons can also be merged (since by applying a positive scaling transformation, the neurons could be brought into the remit of reducibility condition (iii)).
> > * **Algorithms:** The various algorithms could incorporate this new condition by normalising all neuron incoming weights, for example scaling them until the incoming weight has magnitude 1.
> > * **Hardness:** For the hardness result, it becomes necessary to *avoid* an additional reducibility condition in the construction. We would have to carefully check this claim carefully but it seems after initial consideration that the hardness result will hold by essentially the same proof techniques, possibly requiring mild additional assumptions such as assuming there are at least two input units (i.e. the neural network domain is $\mathbb{R}^2$---the effect of introducing the additional degree of freedom per unit being to make up for the positive linear scaling effectively removing one degree of freedom per unit for the construction).
> >
> > Finally we note that, while we are not aware of any other reducibility conditions present in the case of ReLU activation, we don't have to hand a proof that the original three conditions plus this fourth condition are exhaustive. If there are other conditions these would have to be incorporated in turn.
> >
> > A broader comment on the generalisation: While ReLU activation is of course commonly used in practice we note that in future work it should be possible to describe algorithms for lossless compression that take into account affine symmetries of general classes of activation functions including ReLU and others.
> > There is some prior work cited in our paper that goes in this direction in the study of irreducible functional equivalence. For example, see Kurkova and Kainen (1994) and Vlacic and Bolcskei (2021, 2022).

---

### Decision · Program_Chairs · 2023-09-21

**Decision:**

Reject

**Comment:**

The paper handles an important and timely problem and presents interesting theoretical results. The reviewers have raised some issues, and the authors' response has definitely addressed several of those, leaving a few issues open.

Some points that do stand out (these are essentially in the review and discussion too):

• Most of work until Section 5 is reasonable, and Section 6 indeed contains something that looks cool. But given this is a negative result, one key question that one has to ask and address is: “so what should we do now?”
• A point that was raised again and again (and rebutted in a de facto manner, albeit not as deeply as would be useful): why do we care about “lossless” compression of NNs, as well as proximity to such parameters in the first place? (Even if we were to ignore the fact that almost all the data that one feeds into a neural network is almost always noisy and far from perfect.) For tanh networks, the conditions look very stringent and the author should definitely update the work with a discussion (from their rebuttal and interaction with reviewers, and more) about possibly practical implications of studying this problem, even if we were to study epsilon-compressible parameters. The work seems good, but it is definitely missing strongly on a good, solid, motivation (ultimately, we agree that theory will be quite far from practice, especially in a first cut, but it should have carefully argued motivation and relevance.)